# Host Synthesized Carbohydrate Antigens on Viral Glycoproteins as “Achilles’ Heel” of Viruses Contributing to Anti-Viral Immune Protection

**DOI:** 10.3390/ijms21186702

**Published:** 2020-09-13

**Authors:** Uri Galili

**Affiliations:** Rush Medical College, Rush University, Chicago, IL 60612, USA; uri.galili@rcn.com; Tel.: +1-312-753-5997

**Keywords:** zoonotic virus, viral epidemics, glycan shield, *α*-gal epitopes, anti-Gal, Neu5Gc, anti-Neu5Gc, blood groups ABO and Bombay, viral vaccines, Covid-19

## Abstract

The glycans on enveloped viruses are synthesized by host-cell machinery. Some of these glycans on zoonotic viruses of mammalian reservoirs are recognized by human natural antibodies that may protect against such viruses. These antibodies are produced mostly against carbohydrate antigens on gastrointestinal bacteria and fortuitously, they bind to carbohydrate antigens synthesized in other mammals, neutralize and destroy viruses presenting these antigens. Two such antibodies are: anti-Gal binding to *α*-gal epitopes synthesized in non-primate mammals, lemurs, and New World monkeys, and anti-N-glycolyl neuraminic acid (anti-Neu5Gc) binding to N-glycolyl-neuraminic acid (Neu5Gc) synthesized in apes, Old World monkeys, and many non-primate mammals. Anti-Gal appeared in Old World primates following accidental inactivation of the *α*1,3galactosyltransferase gene 20–30 million years ago. Anti-Neu5Gc appeared in hominins following the inactivation of the cytidine-monophosphate-N-acetyl-neuraminic acid hydroxylase gene, which led to the loss of Neu5Gc <6 million-years-ago. It is suggested that an epidemic of a lethal virus eliminated ancestral Old World-primates synthesizing *α*-gal epitopes, whereas few mutated offspring lacking *α*-gal epitopes and producing anti-Gal survived because anti-Gal destroyed viruses presenting *α*-gal epitopes, following replication in parental populations. Similarly, anti-Neu5Gc protected few mutated hominins lacking Neu5Gc in lethal virus epidemics that eliminated parental hominins synthesizing Neu5Gc. Since *α*-gal epitopes are presented on many zoonotic viruses it is suggested that vaccines elevating anti-Gal titers may be of protective significance in areas endemic for such zoonotic viruses. This protection would be during the non-primate mammal to human virus transmission, but not in subsequent human to human transmission where the virus presents human glycans. In addition, production of viral vaccines presenting multiple *α*-gal epitopes increases their immunogenicity because of effective anti-Gal-mediated targeting of vaccines to antigen presenting cells for extensive uptake of the vaccine by these cells.

## 1. Introduction

Enveloped viruses have evolved with envelope proteins that have several N-linked glycosylation sites at the amino acid sequon N-X-S/T, i.e., asparagine-any amino acid (except proline)-serine or threonine and in rare cases at asparagine in the sequon asparagine-any amino acid-cysteine. The asparagine (N) in these sequences serves as amino acid to which high mannose carbohydrate chains or carbohydrate chains of the complex-type are linked. These carbohydrate chains (collectively called “glycans”) are synthesized by the host cell glycosylation machinery [1]. Glycans are also found as O-linked on glycoproteins and linked to lipids as well. The presence of multiple glycans on viral envelope glycoproteins suggests that they may contribute to survival of viruses in hosts. The roles of these glycans may vary and include: 1. Formation of a protective hydration layer around the virus due to the hydrophilic nature of glycans. 2. In some viruses, the sialic acid on these glycoproteins confers a negative electrostatic charge that surrounds the virus and prevents nonspecific adhesion to cell membranes because of the electrostatic repulsion by negative charges of sialic acid on cell membrane glycans. 3. Some glycans bind to a variety of cell surface receptors. 4. Glycans act as a shield (called “glycan shield”) that masks various peptide antigens and prevents binding of neutralizing antibodies, thereby decreasing efficacy of the protective immune response mounted by the host [1,2,3,4,5,6,7,8]. However, these glycans may also be regarded as the “Achilles’ Heel” of the virus, contributing to its vulnerability by causing neutralization and destruction of some viruses infecting humans. This neutralization and destruction are mediated by natural antibodies that bind to carbohydrate antigens on the virus glycan shield. Production of these natural antibodies in humans (and other mammals) occurs throughout life without active immunization. This production is the result of the continuous immune response against a wide range of environmental glycans, most of which are presented on the wall of bacteria that colonize the gastrointestinal tract [9,10,11,12,13]. The variety and amount of bacterial carbohydrate antigens that constantly stimulate the human immune system is enormous as there are ~400 strains of bacteria constituting >25% of the fecal material [14,15]. Fortuitously, some of the immunogenic bacterial carbohydrate antigens have structures that are similar to major carbohydrate antigens on other mammals (e.g., the *α*-gal epitope and N-glycolyl-neuraminic acid (Neu5Gc) antigens), resulting in production of natural anti-carbohydrate antibodies that cross-react with these carbohydrate antigens. Thus, a virus replicating in any of these non-human hosts presents carbohydrate antigens that are synthesized by the host glycosylation machinery and which bind human natural anti-carbohydrate antibodies that may neutralize and destroy the virus. In some cases, the protection by these antibodies may result in complete destruction of the virus and the infected individual would be asymptomatic. In other cases, the protection may be partial. If the neutralization and destruction of the invading virus by the natural anti-carbohydrate antibodies is not complete, there is a “race” within the infected individual between the replicating virus damaging various tissues, and the anti-virus specific elicited immune response, which attempts to prevent the viruses from reaching a lethal mass. In some infected individuals, even partial protection by the natural anti-carbohydrate antibodies may enable the immune system to “catch up” with the expanding virus population and prevent progression to a lethal stage. However, in epidemics of viruses with very high virulence, penetration of even few virions into human cells may suffice to result in replication of virus that has a glycan shield with human glycans, which do not bind any human natural anti-carbohydrate antibody. In this scenario, the subsequent rapid replication of the virus results in the death of a large proportion of the infected hosts, before a protective immune response can be elicited.

This review describes viral carbohydrate antigens that are known to be of great significance in protecting against infections of humans by zoonotic viruses and against infections of blood type O individuals by viruses produced in other blood type individuals (Table 1). The review further describes plausible scenarios in which natural anti-carbohydrate antibodies prevented complete extinction of ancestors of Old World primates and of humans during epidemics that brought these populations to the brink of extinction. The review also describes a future plausible scenario in which humans of extremely rare blood type (called blood group Bombay individuals, or here “Bombay individuals”) may be protected by their natural anti-carbohydrate antibodies against infectious viruses presenting blood group O, A and B antigens, produced in all other humans. In its last part, the review describes methods for increasing the immunogenicity of viral vaccines by glycoengineering their glycan shield in a way that enables harnessing of natural anti-carbohydrate antibodies for targeting viral vaccines to antigen presenting cells (APC), thereby amplifying the efficacy of such vaccines.

## 2. Zoonotic Viral Carbohydrate Antigens and the Human Natural Anti-Carbohydrate Antibodies Binding to Them

Zoonotic viruses are viruses that naturally are found in various mammalian reservoirs, as well as in other animals, and which may be highly virulent upon infection of humans, causing outbreaks, epidemics and pandemics. Such are the virus SARS-CoV-2 causing the current Covid-19 pandemic, influenza virus, and HIV. The carbohydrate antigens on the glycan shields of zoonotic viruses are synthesized by their host. There are two main human natural antibodies known to bind to carbohydrate antigens on mammalian zoonotic viruses and to contribute to protection against infections by such viruses. These are the natural anti-Gal antibody binding to *α*-gal epitopes and anti-N-glycolyl neuraminic acid (anti-Neu5Gc) binding to Neu5Gc (Table 1 and Figure 1). 

Biosynthesis of these carbohydrate antigens on viral glycoproteins within host cells and their subsequent binding of the corresponding human natural antibodies are schematically illustrated in Figure 1. The extent of the ongoing contribution of these antibodies to the continuous protection against various zoonotic viruses in humans is unknown. However, as described in Section 3.3 and Section 5.1, in vitro studies on the effects of these antibodies on such viruses may provide some insight into their ongoing protective activity. The prevention of complete extinction of ancestral Old World primates, described in Section 4 and of hominins in Section 5.2, further support the assumption that these anti-carbohydrate antibodies provide an ongoing protective activity against a variety of zoonotic enveloped viruses.

## 3. The *α*-gal Epitope and the Natural Anti-Gal Antibody 

### 3.1. The *α*-gal Epitope 

One of the major cell surface differences between human cells and non-primate mammalian cells is the expression of *α*-gal epitopes on cell surface glycans. The *α*-gal epitope with the structure Gal*α*1-3Gal*β*1-4GlcNAc-R is synthesized as ~10^5^–10^7^ epitopes/cell on cells of mammals that are not monkeys or apes (non-primate mammals), as well as in lemurs (prosimians that evolved in Madagascar) and in monkeys of South America (New World monkeys) [17,18,19]. However, monkeys of Asia and Africa (Old World monkeys), apes and humans do not synthesize the *α*-gal epitope. The *α*-gal epitope is synthesized in non-primate mammals, lemurs and New World monkeys by the enzyme *α*1,3galactosyltransferase (*α1,3GT*), according to the reaction illustrated in Figure 1A [18,19,20]. Presentation of multiple *α*-gal epitopes on marsupial and placental mammalian cells and the absence of this epitope in cells of fish, amphibians, reptiles and birds [17,18] implies that the *α1,3GT* gene (also called *GGTA1*) appeared early in mammalian evolution before marsupial and placental mammals diverged from a common ancestor (i.e., >125 million years ago [mya]).

### 3.2. The Natural Anti-Gal Antibody

All mammals synthesizing the *α*-gal epitope are immunotolerant to it and cannot produce any antibody against this antigen. However, Old World monkeys, apes and humans lack the *α*-gal epitope and produce a natural antibody that specifically binds to *α*-gal epitopes. This antibody is called the natural anti-Gal (also called anti-*α*-galactosyl, or anti-Gal*α*1-3Gal) antibody [21,22,23,24]. In humans, anti-Gal constitutes as much as 1% of circulating immunoglobulins [21,25,26,27,28]. It is found in the serum as IgG, IgM and IgA and in secretions such as milk, colostrum, bile and saliva as IgA and IgG [21,25,26,27,28]. Anti-Gal IgM is very effective in complement activation and thus in lysis of cells, bacteria and viruses presenting *α*-gal epitopes and in generating complement chemotactic factors. The IgG isotype opsonizes microbial agents and thereby it effectively mediates their phagocytosis and endocytosis by macrophages, dendritic cells and granulocytes. Secretory anti-Gal IgA primarily protects against a variety of microbial agents on mucosal tissues such as the respiratory and gastrointestinal tracts, e.g., neutralization of the influenza virus that may damage the respiratory tract. In a small proportion of humans, anti-Gal is also found as an IgE antibody, which mediates allergic reaction to meat containing *α*-gal epitopes [29,30,31,32]. This allergy is referred to as “*α*-gal syndrome” [30,31,32]. 

Anti-Gal is continuously produced in humans as a result of constant antigenic stimulation by gastrointestinal bacteria that express carbohydrate antigens with structures similar to that of the *α*-gal epitope. This is implied from studies demonstrating binding of anti-Gal to a variety of bacteria found in the gastrointestinal tract, such as some strains of Gram-negative *Escherichia coli* (*E. coli*), *Serratia* and *Klebsiella* and Gram-positive *Enterococcus* and *Streptococcus* [33,34]. The feeding of mice capable of producing anti-Gal (i.e., knockout mice for the *α*1,3galactosyltransferase gene [GT-KO mice]) with such bacteria results in stimulation of their immune system to produce cytolytic anti-Gal antibodies [35]. In accord with these observations, destruction of the gastrointestinal flora in baboons (Old World monkeys) results in a marked decrease in natural anti-Gal production [36]. Glycans containing Gal*α*1-3Glc and Gal*α*1-3Gal structures have been reported in both Gram-positive and Gram-negative bacteria and may comprise part of the multiple galactose containing antigens that stimulate the constant production of anti-Gal within the lymphoid tissues along the gastrointestinal tract [37,38]. 

Anti-Gal is found in the fetal blood as an IgG antibody that crosses the placenta from the maternal blood [21]. This maternal anti-Gal IgG disappears within 3–6 months and is replaced gradually by the newborn’s anti-Gal, which is likely to be induced by the microbial flora established in the newborn gastrointestinal tract. Anti-Gal production gradually increases in the child, reaching the adult level by the age of 2–4 y [21]. Studies in anti-Gal producing mice (knockout mice for the *α1,3GT* gene) have indicated that, similar to the immune response to other bacterial polysaccharides [39,40], B cells produce anti-Gal IgM in response to immunization with *α*-gal epitopes on glycolipids [41]. Isotype switch to anti-Gal IgG production requires activation of helper T cells. However, the *α*-gal epitope, similar to many other carbohydrate antigens on polysaccharides and on N-linked glycans, cannot activate T cells [42]. The T cell help required for isotype switch is provided by helper T cells activated by immunogenic proteins near anti-Gal producing B cells [41,42]. 

Analysis of in vitro anti-Gal secretion by human B cells immortalized by Epstein–Barr virus (EBV) indicated that ~1% of circulating B cells are capable of producing this antibody [43]. In vivo, most of these B cells are quiescent; however, upon administration of xenogeneic cells presenting *α*-gal epitopes into humans or monkeys, these quiescent B cells are activated to proliferate and secrete anti-Gal, thereby increasing this antibody titer >50 fold within a period of 14 days to few weeks [44,45,46]. Anti-Gal binds in vivo very effectively to cell surface *α*-gal epitopes as indicated in xenotransplantation studies (i.e., transplantation of xenogeneic cells and organs from other species [in particular pigs] into monkeys and humans) [47,48,49,50]. This was shown in Old World monkeys, all of which lack the *α*-gal epitope and produce the natural anti-Gal antibody [17,18]. Transplantation of porcine organs (xenografts) into anti-Gal producing monkeys results in rapid binding of anti-Gal within the recipient’s blood to *α*-gal epitopes on the xenograft blood vessel endothelial cells, activation of the complement system, causing complement-dependent cytolysis of these cells, collapse of the vascular bed and rejection of the graft within 30 min to several hours [51,52,53]. As discussed below (Section 3.3), anti-Gal also mediates complement-dependent virolysis of viruses that present *α*-gal epitopes on their envelope glycoproteins by a mechanism similar to that of xenograft cytolysis by this natural antibody. 

### 3.3. Anti-Gal-Mediated Destruction of Enveloped Viruses Presenting *α*-gal Epitopes

Viral envelope glycoproteins with glycans of the complex-type undergo glycosylation, which starts in the endoplasmic reticulum and is completed within the Golgi apparatus by the host cell glycosyltransferases, similar to glycosylation of cellular glycans [1,54]. Figure 1A illustrates the last step (also called “capping”) in the synthesis of *α*-gal epitopes. The first example of differential glycosylation on a given virus, due to the presence or absence of *α1,3GT* in the host cell, was the synthesis of *α*-gal epitopes on Eastern equine encephalitis virus (EEEV) replicating in mouse and in monkey cells. Replication of the virus in mouse 3T3 fibroblasts resulted in synthesis of *α*-gal epitopes on EEEV, whereas replication of EEEV in Vero cells (African green monkey-Old World monkey) resulted in production of virions that lack this epitope [55]. Similarly, influenza virus produced in Madin–Darby Bovine Kidney (MDBK) cells or in Madin–Darby Canine Kidney (MDCK) cells (both having *α*1,3GT) presents *α*-gal epitopes, whereas the same influenza virus replicating in chicken cells (birds have no *α*1,3GT) lacks these epitopes [56]. *α*-gal epitopes were further identified on viruses replicating in mammalian cells containing active *α1,3GT*, including: Friend murine leukemia virus replicating in mouse cells [57,58], porcine endogenous retrovirus (PERV) and pseudo-rabies virus replicating in porcine cells [59,60], rhabdo-, lenti-, and spumaviruses replicating in murine, hamster and mink cells [61], Newcastle disease virus, Sindbis virus and vesicular stomatitis virus (VSV) replicating in murine and hamster cells [62,63], measles virus [64,65] and vaccinia virus [66] replicating in human cells that were transfected with the *α1,3GT* gene and influenza virus engineered to contain the *α*1,3GT gene and which replicated in embryonated chicken eggs [67]. Viruses presenting *α*-gal epitopes were found to be lysed by anti-Gal and complement in human serum whereas same viruses lacking *α*-gal epitopes were not lysed when incubated in human serum [58,59,60,61,62,63,64,65,66]. An example for the extent of such virolysis is that of VSV presenting *α*-gal epitopes which undergoes 99.9% lysis following incubation for 45 min in 50% human serum [63]. All these virolysis studies suggest that a similar virolytic mechanism occurs in vivo where anti-Gal functions as a natural immune barrier, which prevents infections by zoonotic viruses originating in non-primate mammalian reservoirs. Transmission of such viruses to humans may be caused by bites from such mammals; aerosolized, fecal, or urinary secretions from these mammals; or by eating them. 

The protection against infection of humans by zoonotic viruses presenting *α*-gal epitopes is not absolute as can be inferred from the SARS-CoV-2 virus transmission from bats (mammals synthesizing *α*-gal epitopes [18]), or from the outbreaks of infection by influenza virus replicating in pigs followed by transmission to humans. Moreover, some of the virions evade destruction by anti-Gal and “succeed” in infecting human cells. Replication within the human cells results in the production of a virus lacking *α*-gal epitopes on their glycoproteins. Nevertheless, even partial anti-Gal-mediated destruction of the invading virus presenting *α*-gal epitopes and the effective anti-Gal-mediated targeting of the destroyed virus to APC (described in Section 3.4) may contribute to the amplification of the specific anti-virus elicited immune response, and the prevention of the virus from reaching a lethal mass. As further suggested in Section 4, these protective effects of anti-Gal might have prevented the complete extinction of ancestral Old World primates (monkeys and apes). 

An indirect example for anti-Gal-mediated protection against zoonotic viruses is the study on PERV infection in humans exposed to live porcine cells and tissues [68]. The observation that PERV, which is found in pigs [59], can also infect human cells [69] raised the concern of cross-species transmission of PERV in future recipients of porcine xenografts (e.g., porcine heart or kidney) who may be infected by this virus. Thus, studies have been performed on detection of PERV genome (by RT-PCR) in mononuclear cells of 160 patients that were treated with various live pig tissues. Among the patients studied were those treated with pig skin for covering burns, patients suffering from infection whose blood was perfused through pig spleen and diabetic patients transplanted with fetal pig islet cells. No PERV genome was detected in any of the patients studied, indirectly suggesting that the virus within the porcine tissue xenograft did not cross the anti-Gal barrier for infecting human cells in these xenograft recipients [68]. Taken together with the in vitro observations on anti-Gal-mediated virolysis of PERV propagated in porcine cells or in human cells engineered to produce *α1,3GT* [59], these findings [68] support the assumption that anti-Gal can lyse viruses presenting *α*-gal epitopes, upon infection of humans. The efficacy of anti-Gal-mediated protection against zoonotic viruses is likely to vary from one virus strain to the other and depends on the virulence of each strain and the number of *α*-gal epitopes on each virion. The titer and avidity of anti-Gal in individual patients and the efficacy of the immune system of various patients in mounting protective specific anti-virus T cell and B cell immune response are likely to determine the ultimate outcome of the zoonotic virus infection as well. 

### 3.4. Increased Specific Anti-Viral Immune Response by Anti-Gal-Mediated Targeting to APC

As indicated above, it is possible that despite the effective destruction of invading zoonotic viruses by anti-Gal, some of the virions will “succeed” in infecting cells in which they will replicate without presentation of *α*-gal epitopes. Under such circumstances, prevention of the virus from reaching a lethal mass depends on the ability of the immune system to generate a sufficient number of virus specific T cells that kill virus infected host cells and sufficient titer of virus specific antibodies that bind to virus envelope proteins and effectively neutralize and destroy the virus budding from infected cells. Evidently, in this “race” between the virus replicating toward reaching a lethal mass and the anti-virus specific immune response thwarting progression of the infection, shortening the period required for reaching an effective level of protective immune response will increase the probability of overcoming the viral infection. 

The activation of virus specific helper T cells and cytolytic T cells (CTL-killing cells infected with the replicating virus) requires the uptake of the lysed virus within endosomes of APC such as dendritic cells and macrophages. The APC transport the internalized virus to the draining lymph nodes and spleen. In the endosomes, the APC further destroy the viral glycans by glycosidases and degrade the viral proteins by proteolysis into antigenic peptides that are presented by cell surface major histocompatibility complex (MHC) molecules. The presented peptides engage the corresponding T cell receptors, resulting in the activation and proliferation of these T cells. The expanding activated virus specific CTL circulate and destroy cells infected with the virus whereas the helper T cells assist virus specific B cells to produce the corresponding antibodies that neutralize and destroy viruses released from the infected cells. In general, the uptake of vaccinating antigens by APC is of limited efficacy and is mediated by random endocytosis of virus or of protein molecules that are accidently present near the surface of the APC. However, zoonotic viruses lysed by anti-Gal are effectively targeted to APC and are internalized by these cells. The reason is that the Fc portion of anti-Gal immunocomplexed with the lysed virus binds effectively to Fc receptors on the cell membrane of the APC (Figure 2).

The efficacy of this anti-Gal-mediated targeting of virus to APC in amplifying the immune response was demonstrated both in vitro and in vivo. In vitro, uptake by APC and processing of inactivated influenza virus presenting *α*-gal epitopes on the glycan shield (following replication in MDCK canine cells) was found to be ten-fold higher than that of inactivated virus in the absence of human anti-Gal antibody [56]. In contrast, uptake, processing and presentation by APC of influenza virus antigens, when the virus lacked *α*-gal epitopes (replicated in chicken cells), was the same in the presence or absence of human natural anti-Gal antibody. In vivo studies were performed on GT-KO mice [70] producing anti-Gal following immunization with pig kidney membranes homogenate [41,71]. Presentation of multiple *α*-gal epitopes on the glycan shield of immunizing inactivated influenza virus was achieved by enzymatic glycoengineering using recombinant *α1,3GT* and UDP-Gal, similar to the intracellular reaction presented in Figure 1A [71,72]. Immunization of the mice with a vaccine consisting of inactivated influenza virus presenting *α*-gal epitopes demonstrated ~100-fold increase in anti-virus antibody response, ~6-fold increase in CD8^+^ T cell response and ~100-fold increase in CD4^+^ T cell response in comparison to mice immunized with inactivated influenza virus lacking *α*-gal epitopes [71]. Furthermore, the intranasal challenge of the mice with a lethal dose of live influenza virus lacking *α*-gal epitopes was followed by death within 10 days for ~90% of mice immunized with virus lacking *α*-gal epitopes, whereas only ~10% of mice immunized with virus presenting *α*-gal epitopes died and the rest survived without any subsequent indication of infection [71]. A similar amplification of the immune response was observed against gp120 of HIV in anti-Gal producing GT-KO mice immunized with gp120 glycoengineered to present *α*-gal epitopes vs. that in mice immunized with gp120 lacking *α*-gal epitopes [73]. These observations strongly suggest that anti-Gal-mediated targeting of lysed zoonotic viruses to APC is likely to initiate an early anti-viral specific protective immune response, which is more effective than the immune response elicited in the absence of immunocomplexing. This amplification of immune response was also observed with bovine albumin [74] and ovalbumin [75] immunocomplexed with anti-Gal at the vaccination site. A recent study using influenza virus with inserted *α1,3GT* gene, demonstrated de novo presentation of *α*-gal epitopes on cells infected with the virus [67]. Vaccination of anti-Gal producing GT-KO mice with this attenuated influenza virus also demonstrated elevated resistance to live influenza virus [67]. It would be of interest to determine whether the influenza virus produced in cells infected with this vaccinating virus also present *α*-gal epitopes. 

The natural anti-Gal antibody activity in various individuals was reported to be at a wide range [21,27,76,77]. Therefore, the observations on the protective effects of anti-Gal against zoonotic viruses further suggest that individuals travelling to regions suspected to have high concentrations of such viruses replicating in non-primate mammalian reservoirs may benefit from receiving vaccines that elevate the titer of this antibody (e.g., *α*-gal linked to a xenoprotein) [16,78], thereby improving the immune protection against zoonotic viruses. The observations on complement-dependent cytolytic activity of anti-Gal on a variety of protozoa [79], including *Trypanosoma cruzi* [80], *Leishmania major* [81] and *Plasmodium* spp. [82] have led to similar suggestions that increasing anti-Gal titers in individuals residing in or traveling to areas endemic for these parasites may have protective effects against parasites presenting *α*-gal epitopes. 

## 4. Proposed Anti-Gal-Mediated Protection from Extinction of Old World Primates in Viral Epidemic

The *α*-gal epitope is synthesized in cells of non-primate mammals, lemurs and New World monkeys and thus is synthesized on the glycan shield of enveloped viruses replicating in these cells, but is completely absent in Old World monkeys, apes, and humans. Since the *α*-gal epitope is synthesized only in mammals (both marsupials and placentals) it is probable that it appeared in early mammals, prior to divergence of marsupials and placentals >125 mya [17,18]. However, the *α*-gal epitope was eliminated in Old World primates (monkeys and apes) as a result of a selective evolutionary process among ancestral primates that lived in the Eurasia–Africa landmass and not in other geographically isolated areas (i.e., the island of Madagascar and the continent of South America) [17,18]. Studies aimed to understand the molecular basis for the *α*-gal epitope elimination in humans and Old World primates have demonstrated the presence of an *α1,3GT* pseudogene with one or very few base deletions that result in pre-mature stop codons that completely inactivate the catalytic activity of the produced *α1,3GT* enzyme [83,84,85,86,87]. Humans and apes (chimpanzee, gorilla and orangutan) display a conserved inactivating base deletion absent in Old World monkeys [84]. The DNA sequence of the pseudogene in Old World primates suggests that the selective process resulting in survival of primates that had the inactivated *α1,3GT* pseudogene occurred in the Eurasia–Africa landmass after the divergence from New World monkeys, ~20–30 mya (Figure 3).

Taken together, the conserved base deletion of the *α1,3GT* pseudogene in apes [84], the production of anti-Gal in all Old World primates [17,88] and the anti-Gal-mediated destruction of enveloped viruses replicating in mammals other than Old World primates (58–66), suggest a unique evolutionary event. All these observations raise the possibility that the elimination of primates synthesizing the *α*-gal epitope and the survival of primate progeny lacking this epitope and producing the natural anti-Gal antibody was the result of a “catastrophic selection” process. This process could have been mediated by an epidemic of a lethal virus that occurred in the Eurasia–Africa landmass, in a process analogous to present-day protection by anti-Gal against zoonotic viruses [89]. An evolutionary “catastrophic selection” process is defined as: “…an entire population is eliminated by an environmental extreme, except for one or more exceptionally adapted individuals… characterized by a deviant genome. By elimination of the parental population, catastrophic selection isolates the survivors in an open habitat to which they are adapted” [90]. Primates in South America and lemurs in Madagascar were not exposed to this epidemic because they were protected by oceanic barriers.

As illustrated in Figure 4, it is suggested that this selection process occurred among early Old World primates. These primates that lived after the evolutionary divergence from New World monkeys (30–40 mya) had active *α1,3GT* that synthesized *α*-gal epitopes, as New World monkeys and non-primate mammals. An accidental base deletion mutation inactivating the enzyme coded by the mutated *α1,3GT* gene (*GGTA1*) occurred in one individual in one of the two alleles of the gene. In this individual, synthesis of *α*-gal epitopes continued by the enzyme produced by the second non-mutated allele. After several generations, a number of individuals in that ancestral primate population were homozygous for the mutated gene, thus they could not synthesize the *α*-gal epitope. In absence of the *α*-gal epitope, such progeny lost immune tolerance to it and produced the natural anti-Gal antibody as a result of antigenic stimulation by gastrointestinal bacteria. An epidemic caused by a lethal enveloped virus killed the parental population synthesizing *α*-gal epitopes. The lethal virus replicating in the parental population presented *α*-gal epitopes on envelope glycoproteins. Upon infection of offspring lacking *α*-gal epitopes and producing anti-Gal, the virus was prevented from reaching lethal mass by this antibody, resulting in survival of such offspring. Ultimately, this selection process resulted in the complete replacement of the *α*-gal epitope producing Old World primates with offspring lacking *α*-gal epitopes and producing the natural anti-Gal antibody.

The hypothesis illustrated in Figure 4 is supported by several observations including: 1 the one base deletion conserved in apes and humans (base G904) [84,89] suggests that this selection process occurred in one early ancestral Old World ape population. If it had occurred independently in several different epidemics in the course of ape evolution, there would have been different inactivating mutations in different ape species. The earliest mutation inactivating the *α1,3GT* gene in Old World monkeys and possibly both in monkeys and apes awaits further studies on the full sequence of this pseudogene in a sufficient number of Old World monkeys and apes. 2. The production of the natural anti-Gal antibody is expected to have initiated in the first primates that were homozygous for the inactivated *α1,3GT* gene. Production of anti-Gal against the lost *α*-gal epitope indeed can be presently observed in studies on *α1,3GT* knockout pigs [91,92]. Homozygous progeny for the inactivated allele, lacking *α*-gal epitopes, were generated from mating of two heterozygous pigs, each with only one inactivated allele. These homozygous progeny produce the natural anti-Gal antibody whereas the parental heterozygotes synthesize the *α*-gal epitope and do not produce anti-Gal [93,94,95]. This implies that although the porcine evolutionary lineage never produced anti-Gal for >100 million years, throughout mammalian evolution, it produced this natural antibody once the *α*-gal epitope was eliminated in these animals. 3. The assumption that among ancestral Old World primates synthesizing *α*-gal epitopes there was a very small population with inactivated *α1,3GT* gene that lacked *α*-gal epitopes and produced the natural anti-Gal antibody has a present day analogous example among humans. This are the blood group Bombay individuals (called here “Bombay individuals”) who lack blood group O (also called the H antigen) due to the lack of *α*1,2 fucosyltransferases and produce natural anti-H (anti-blood group O) antibody [96,97,98]. There are ~1:10,000 Bombay individuals in India and 1:300,000 to 1:1,000,000 in other countries [99]. This blood group, the mutations resulting in elimination of the H antigen and the implications for protection against viruses presenting the H antigen, are further discussed in Section 7. The very few Bombay individuals demonstrate a present day scenario of a few mutated individuals who lost a common carbohydrate antigen (i.e., the H antigen) and produce a natural anti-H antibody against the lost antigen. This scenario is analogous to the few primates lacking *α*-gal epitopes and producing anti-Gal, which existed prior to the suggested catastrophic selection process that eliminated ancestral Old World primates synthesizing the *α*-gal epitope. 

## 5. Anti-Neu5Gc Protecting against Viruses Presenting Neu5Gc

### 5.1. Distribution of Neu5Gc and Lysis of Viruses Presenting it

A second carbohydrate antigen that may be found on mammalian zoonotic viruses and which is recognized by a human natural antibody is N-5-glycolyl-neuraminic acid (Neu5Gc), which is a sialic (neuraminic) acid antigen. As illustrated in Figure 1B, Neu5Gc provided by the sugar donor cytidine-monophosphate-Neu5Gc (CMP-Neu5Gc) is linked by sialyltransferases to nascent carbohydrate chains on glycans of cells and of viral envelope glycoproteins. In Old World monkeys and apes (as in many non-primate mammals) Neu5Gc is synthesized from N-acetyl-5-neuraminic acid (Neu5Ac) by adding a hydroxyl that converts the N-acetyl into N-glycolyl. This hydroxylation is the result of catalytic activity of the enzyme cytidine-monophosphate-N-acetyl-neuraminic acid hydroxylase (CMAH) [100]. Whereas glycans present both Neu5Ac and Neu5Gc on cells of non-primate mammals, Old World monkeys and apes (including chimpanzee), humans synthesize only Neu5Ac and lack Neu5Gc because the *CMAH* gene is inactive in humans [101,102]. This inactivation is the result of a 92 bp deletion that forms a premature stop codon which truncates the *CMAH* product, thereby preventing activity of CMAH [103,104]. In analogy to humans who lack the *α*-gal epitope and produce the natural anti-Gal antibody against it, humans lack Neu5Gc and produce natural anti-Neu5Gc antibody against this antigen [101,102,105,106,107,108]. An earlier name for this antibody has been Hanganutziu–Deicher antibody [105]. 

The synthesis of Neu5Gc in many mammalian and primate cells vs. the absence of this antigen and production of natural anti-Neu5Gc antibody in humans raised the possibility that multiple zoonotic viruses replicating in various mammalian reservoirs are likely to present Neu5Gc on their envelope glycans and that anti-Neu5Gc antibody may serve as an immune barrier against infections by such viruses [63,109]. Recent studies indeed demonstrated complement-dependent virolysis of vesicular stomatitis virus (VSV) presenting Neu5Gc (i.e., replicating in African Green monkey Vero cells) by natural anti-Neu5Gc antibody within human serum [63]. Studies on VSV replicating in mouse 3T3 cells and thus presenting both *α*-gal epitopes and Neu5Gc suggest that the natural anti-Gal antibody in human serum is approximately ~10-fold more potent in inducing complement-dependent virolysis than anti-Neu5Gc antibody [63]. Additional research is required for establishing the significance of anti-Neu5Gc in the protection against zoonotic viruses. However, these observations suggest that analogous to the suggestion above for immunization that elevates anti-Gal titers in travelers to areas with various non-primate mammalian zoonotic viruses [16,78], immunization for elevating anti-Neu5Gc antibody titers may be beneficial in protection against zoonotic viruses replicating in Old World monkey and ape reservoirs. This suggestion for the feasibility for such vaccination is supported by observations demonstrating increased titers of anti-Neu5Gc antibody in patients exposed to Neu5Gc on porcine glycans [110].

### 5.2. Suggested Significance of Viral Neu5Gc Interaction with Anti-Neu5Gc Antibody in Human Evolution

The synthesis of both Neu5Ac and Neu5Gc in all apes and Old World monkey tested and the complete absence of Neu5Gc in humans implies that the elimination of Neu5Gc in ancestors of humans (called hominins) occurred after they and ancestors of chimpanzee diverged from common ancestral ape, i.e., <6 mya [101,102,109,111]. Thus, it is reasonable to assume that early hominins synthesized both Neu5Ac and Neu5Gc, as presently observed in chimpanzee and other apes. In view of the observed production of anti-Neu5Gc antibody in humans [101,102,105,106,107,108], it may be possible that the evolutionary scenario that led to the loss of Neu5Gc had similar “catastrophic selection” characteristics as those described above for the elimination of *α*-gal epitopes and appearance of the natural anti-Gal antibody in ancestral Old World monkeys and apes (illustrated in Figure 4). In this scenario there were very few individuals among the early hominins who were accidentally homozygous for the mutation inactivating the *CMAH* gene and thus, they did not synthesize Neu5Gc. Upon the loss of Neu5Gc, these few individuals ceased to be immunotolerant to it and produced the natural anti-Neu5Gc antibody. It is suggested that early hominins were exposed to an epidemic of a lethal virus. The virus replicating in the parental hominin populations presented on its glycans both Neu5Ac and Neu5Gc synthesized by the host glycosylation machinery. Whereas infection of the parental hominins resulted in their death and extinction, infection of the offspring lacking Neu5Gc resulted in the destruction of the virus by the natural anti-Neu5Gc antibody they produced [89]. This suggested scenario is in line with the observation on the ability of the natural anti-Neu5Gc antibody in human serum to induce complement-dependent virolysis of 99.9% of VSV presenting Neu5Gc following replication in Vero cells (Old World monkey cells), whereas no such virolysis is observed in VSV replicating in human cells [63]. 

In addition to its absence in humans, Neu5Gc was found to be absent in some mammalian groups such as New World monkeys, sperm whales, ferrets and white-tailed deer [109]. These observations may suggest that the scenario of catastrophic selection by elimination of Neu5Gc and the ensuing production of natural anti-Neu5Gc antibody, as that in hominins, also occurred in the course of evolution of several mammalian species. The absence of Neu5Gc in these species further suggests that Neu5Gc is not an essential molecule and may be replaced by Neu5Ac. It is possible that for this reason, accidental elimination of Neu5Gc enabled the production of the anti-Neu5Gc antibody, which prevented the complete extinction of various mammalian species that were at the brink of extinction in epidemics caused by lethal viruses, analogous to the catastrophic selection in ancestral hominins. 

## 6. Immune Protection against Viruses Presenting Blood Group A and B Antigens

### 6.1. Enveloped Viruses Replicating in Human Cells Present ABO Antigens of the Host and Are Lysed In Vitro by the Corresponding Anti-Blood Group Antibodies 

Some insight into the extent of immune protection by natural anti-carbohydrate antibodies in humans may be gained from studies on susceptibility of individuals of various ABO blood groups to infection by enveloped viruses transmitted among humans cf. [112]. Enveloped viruses replicating in human cells present the blood group O antigen (called H antigen) with the structure Fucα1-2Galβ1-4GlcNAc-R. This antigen is synthesized in all humans (except for blood group Bombay individuals) by *α*1,2fucosyltransferases (*α1,2FT*), which link fucose provided by GDP-Fuc to Galβ1-4GlcNAc of the nascent carbohydrate chain (Figure 5). In blood group A individuals, the A transferase links terminal GalNAcα1-3 to the penultimate galactose of a portion of the H antigen molecules to synthesize the A antigen. Therefore, virus replicated in blood group A host cells will present primarily A antigen (Figure 5), and also H antigen on some of the glycans that were not capped by the A transferase [64]. Similarly, virus replicating in blood group B host cells will present primarily the B antigen (Figure 5) and also the H antigen on glycans that were not capped by the B transferase.

The differential blood group antigen presentation on viruses, according to the blood group transferases in human host cells, was demonstrated with measles virus and HIV replicating in human cells [64,65,113,114]. These studies also demonstrated lysis of viruses presenting blood group A by anti-A antibody in human serum and those presenting blood group B, by anti-B antibody. Moreover, in SARS-CoV virus presenting blood group A on the S protein, adhesion of the virus to its receptor on target cells was inhibited following binding of anti-blood group A antibodies to the S protein [115]. This suggests that anti-blood group A antibodies may block the interaction between the virus and its receptor, thereby providing protection even in the absence of complement-dependent virolysis. The protective effect of human anti-blood group A or B antibodies against viruses presenting the corresponding carbohydrate antigen is hard to evaluate in vivo. The section below attempts to address this issue, indirectly. However, it is of note that these antibodies are very potent in in vivo activation of the complement system. This is evident in rare cases of ABO mismatched red blood cells provided to patients by transfusion, which result, within minutes, in extensive hemolysis of the transfused red blood cells. If not treated rapidly, this reaction can be life threatening.

### 6.2. Anti-A/B Antibody Protection in Humans against Infection by Viruses Presenting Blood Groups A or B Antigens

The in vitro virolysis of enveloped viruses presenting blood group A or B antigen by anti-A or anti-B antibody, respectively, as described above, suggests that individuals who are of blood group O will display less susceptibility to infection by viruses that present blood group A or B antigens than those of blood group A, B or AB. A famous example supporting this suggestion is the report on a blood group A patient with severe acute respiratory syndrome (SARS) that infected unprotected hospital workers with SARS-CoV during the 2002–2003 SARS outbreak in Hong Kong [116]. The least susceptible to infection among the symptomatic infected individuals were of blood group O. Also in the current Covid-19 pandemic, the proportion of blood group O individuals among the symptomatic patients was reported to be significantly lower than their proportion in the general population, whereas that of blood group A patients was significantly higher than that in the general population [117,118,119]. Although these studies require further substantiation by performing analysis on larger patient populations, these preliminary observations support the assumption that anti-blood group A and B antibodies may be potent enough in a proportion of various populations to destroy invading viruses presenting the corresponding carbohydrate antigens.

### 6.3. Possible Viral Cause for the Exclusive Blood Group O among Indians of the Amazon

In most countries, human populations include all four blood groups A, B, AB and O [120], with the exception of the Indians in the Amazon basin where 100% of individuals are of blood group O [121,122]. It is thought that South America Indians originated from North America Indians that migrated South across the land bridge between the two continents, 10,000–20,000 years ago. Blood group A is prevalent among North America Indians (20–50% of population in various tribes) [123,124], strongly suggesting that this blood group was likely to be present in Indian populations migrating to South America and was eliminated among South America Indians in a period of <20,000 years. Indeed, blood group A was found on tissues of <3000 year old pre-Columbian Chilean and Peruvian mummies [125]. In view of the observations on virolysis of enveloped viruses presenting the blood group A antigen by human anti-A antibody [64,65,113,114] and of the corresponding decreased susceptibility of blood group O individuals to infectious viruses presenting blood group A antigen [116,117,118,119], it is suggested that various deleterious virus outbreaks in South America Indian communities resulted in higher susceptibility of blood group A adults and newborns to viruses replicating in blood group A and O hosts than of blood group O adults and newborns infected by the same virus replicating in blood group A hosts. The natural anti-blood group A antibody produced in blood group O individuals destroyed viruses presenting blood group A antigen, whereas blood group A individuals lacked protective antibodies against viruses that originated in blood group A or O individuals. It may be possible that such repeated outbreaks gradually resulted in complete elimination of blood group A individuals, as also anticipated using a mathematical model [113]. This elimination of blood group A individuals by lethal viruses and the resistance of blood group O individuals due to the production of the natural anti-blood group A antibody that destroys virus transmitted from blood group A individuals may be in line with the scenarios of virus mediated evolutionary elimination of ancestral Old World primates, and of hominins, as suggested above. 

## 7. Blood Group Bombay Individuals and Future Viral Epidemics 

The virolytic effect of the anti-blood group A and B antibodies on viruses presenting the corresponding blood group antigen also enables a suggested extrapolation regarding the future protective role of the natural anti-blood group O (called anti-H antibody) in individuals with the rare blood group Bombay (called here Bombay individuals). The protective effect of the anti-H antibody may be against viruses presenting the H antigen that replicate in >99.99% of humans [96,97,98,99,126,127,128] (Figure 5). This is because some of the glycans on viruses replicating in blood group A, B and AB individuals are not capped by α1-3GalNAc or α1-3Gal and present the H antigen [64]. As mentioned in Section 4, Bombay individuals lack the ability to synthesize the H antigen due to the inactivation of their H-transferases. Bombay individuals are found as 1:10,000 in India and 1:300,000 to 1:1,000,000 in other countries [96,97,98,99]. In a comprehensive study performed in recent years on seven million first-time blood donors in Iran, only 56 individuals were found to be of blood group Bombay [128]. Consanguinity was observed in 50 cases (89%), which is expected because of the extremely low probability of finding non-related individuals carrying mutations inactivating the H-transferase genes.

The H antigen is synthesized by H-transferases, also called *α*1,2fucosyltransferases (*α1,2FT*) (as in Figure 5), which are coded by the *FUT1* gene for the H-transferase synthesizing the H antigen on red blood cells [129] and *FUT2* gene for the H-transferase that synthesizes the H antigen in its secretory form [130,131]. *FUT2* mutations inactivating this gene are found in 20% of humans, whereas inactivating mutations in *FUT1* are extremely rare [132,133,134,135,136,137]. The absence of both these fucosyltransferases in Bombay is the result of inactivating base deletion mutations that are different in various geographic areas [132,133,134,135,136,137], implying that the rare inactivating mutation events in *FUT1* occurred after humans migrated from Africa to other parts of the world. Similar to the appearance of the natural anti-Gal antibody in Old World primates once the *α*-gal epitope was lost, humans homozygous for both inactivated *FUT1* and *FUT2* genes do not synthesize the H antigen and lose the immune tolerance to it. Therefore, they produce the natural anti-H antibody, possibly as a result of the immune response to an antigen with a structure similar to the H antigen, which is presented by bacteria of the natural flora. In the absence of the H antigen, Bombay individuals cannot synthesize blood groups A and B antigens, and thus, they produce the natural anti-A and anti-B antibodies as well. The combination of natural anti-H, anti-A and anti-B antibodies produced in Bombay individuals may contribute to immune protection against enveloped viruses produced in any human host who is not of blood group Bombay (Figure 5). The in vivo potency of the natural anti-H antibody in mediating complement-dependent cytolysis of cells presenting the H antigen was observed in Bombay individuals who were misdiagnosed as blood group O individuals and thus, were transfused with blood group O red blood cells. These red blood cells were lysed within minutes as a result of binding of the anti-H antibody to the H antigen on these cells and the activation of the complement system. This transfusion reaction may be lethal if not stopped in time and the patient treated [97,138,139,140].

In view of this potent activity against the H antigen, one may speculate that in future epidemics of deleterious enveloped viruses that spread by human to human transmission, Bombay individuals may have a first line of immune defense, absent in all other humans. This defense is the combined activity of natural anti-H, anti-A and anti-B antibodies inducing complement-dependent virolysis of viruses replicating in any ABO individual. Thus, Bombay individuals may have an advantageous natural immunity in populations that do not receive timely vaccination against pandemic-causing viruses, due to the production of natural anti-H, in addition to anti-A and anti-B antibodies.

## 8. Glycoengineering Viral Vaccines for Presenting *α*-gal Epitopes that Increase Vaccine Efficacy

The studies described in Section 3.4 support the assumption that *α*-gal epitopes presented on glycan shields of zoonotic viruses increase the immunogenicity of viral proteins by increased uptake into APC, as illustrated in Figure 2 [56,71,72,73,74,75]. Moreover, uptake of vaccinating immune complexes by APC via Fc/Fc receptor interaction induces maturation of these cells into professional APC that effectively process and present the immunogenic peptides of internalized protein antigens, ultimately resulting in amplifying the anti-virus protective immune response [141,142]. This raises the possibility that glycoengineering the glycan shield of vaccinating viruses to present *α*-gal epitopes may convert the original glycan shield from a component of the vaccine that decreases immunogenicity into one that amplifies the immunogenicity of the protein portion of envelope glycoproteins. Such amplified immunogenicity of viral vaccines is of particular significance in aging populations where the protective immune response to these vaccines (e.g., influenza virus vaccines) is weaker than in young populations. Anti-Gal-mediated increased immunogenicity by glycoengineering of glycan shield to present *α*-gal epitopes may also be applicable to prospective SARS-CoV-2 vaccines in the Covid-19 pandemic [143]. Thus, the mechanism, which originally was conceived for increasing immunogenicity by anti-Gal/*α*-gal immunocomplexing for the conversion of autologous tumors engineered to present *α*-gal epitopes into anti-tumor vaccines [144,145,146], is also applicable to amplifying immunogenicity of viral vaccines, as discussed in this review.

It is probable that there is a direct correlation between the number of *α*-gal epitopes per vaccinating virion or per vaccinating envelope glycoprotein molecule (e.g., subunit vaccine or recombinant glycoprotein vaccine) and the extent of anti-Gal-mediated targeting of the vaccine to APC. Thus, developing methods for maximizing the number of *α*-gal epitopes on the glycan shield of viral vaccines will contribute to the optimization of such vaccines. Three of these methods are the following:
Synthesis of *α*-gal epitopes by recombinant *α*1,3galactosyltransferase (*rα1,3GT*): truncated *rα1,3GT* cDNA lacking the cytoplasmic and trans-membrane domains can be cloned from various mammalian cells (e.g., *α1,3GT* cDNA cloned from New World monkey cells [147]) and produced in any expression system. The *rα1,3GT* transfers galactose from the UDP-Gal high energy sugar donor to N-acetyllactosamine (Galβ1-4GlcNAc-R) on desialylated glycan of envelope glycoprotein on inactivated viruses, or on the glycan of an isolated desialylated glycoprotein, similar to the intracellular reaction illustrated in Figure 1A. The reaction requires presence of Mn^++^ ions. When performed with inactivated influenza virus (no desialylation is required with this virus), the reaction resulted in synthesis of ~3000 *α*-gal epitopes per virion, i.e., capping of possibly all glycans of the complex-type with *α*-gal epitopes [72]. Similarly, *rα1,3GT* was found to be effective in synthesis of multiple *α*-gal epitopes on desialylated recombinant gp120 of HIV [73] and on a recombinant fusion protein combined of desialylated gp120 fused to p24 of HIV [148].Intracellular synthesis of *α*-gal epitopes on envelope glycoproteins: the natural synthesis of *α*-gal epitopes on cellular and viral glycoproteins occurs in the trans-Golgi apparatus where the *α1,3GT* competes with other glycosyltransferases, such as sialyltransferases, for capping the nascent glycan with terminal Gal*α*1-3 for formation of the *α*-gal epitope or capping with sialic acid, respectively [149]. Due to this competition, only a portion of the complex-type glycans on a replicating virus or on a recombinant viral glycoprotein will be capped by the *α*-gal epitope (Figure 1A) and the rest will be capped by other carbohydrates, mostly by sialic acid (Figure 1B). In order to maximize the number of synthesized *α*-gal epitopes, the host cells used for propagation of the vaccinating virus should undergo stable transfection with several copies of the full length active *α1,3GT* gene (*GGTA1*). This is likely to increase the concentration of *α1,3GT* in the trans-Golgi compartment. To further decrease the competition between *α1,3GT* and sialyltransferases, thereby increasing capping by the *α*-gal epitope, sialyltransferase genes may be disrupted (i.e., knocked out) in the host cells. An alternative approach for introducing multiple copies of the *α1,3GT* gene is to transduce the host cells with replication-defective adenovirus containing the *α1,3GT* gene (AdαGT) [150] prior to the infection with the replicating virus to be used as the vaccine. This replication-defective adenovirus was found to introduce ~20 copies of the *α1,3GT* gene into HeLa cells. These copies of the *α1,3GT* gene are transcribed by 4 hr post transduction and the synthesized *α*-gal epitopes are detected on the cell membrane within 10 hr post transduction [150]. Thus, transduction of host cells 12–24 hr prior to infection with the vaccinating virus may result in activity of a high concentration of *α1,3GT*, which synthesizes multiple *α*-gal epitopes on the viruses replicating in these cells. In addition, it remains to be determined whether insertion of the *α1,3GT* gene into the virus genome, as demonstrated in ref. [67], results in synthesis of multiple *α*-gal epitopes on the propagated virus and if the production of such a virus is at a high enough yield that it enables the preparation of the inactivated virus vaccines, or subunit and split vaccines needed for large populations.Synthesis of recombinant viral glycoproteins presenting *α*-gal epitopes in glycoengineered yeast and bacteria: recombinant viral envelope glycoproteins that carry *α*-gal epitopes on their glycan shield may be considered as a source for vaccine preparation. Production of large amounts of such glycoproteins may be feasible in yeast. Yeast do not have the glycosylation machinery for synthesis of N-linked glycans of the complex-type, but only of the high mannose-type. In the last two decades, a technology for glycoengineering yeast to synthesize glycans of the complex-type has been developed by introducing into yeast glycosyltransferase genes coding for the enzymes that synthesize these glycans [151,152]. This technology enables the production of therapeutic glycoproteins with “humanized” glycans that are capped with the oligosaccharide of sialic acid linked to N-acetyllactosamine (SA-Galβ1-4GlcNAc-R), characteristic to human glycans. It is suggested that engineering such yeast with the *α1,3GT* gene instead of sialyltransferases genes may result in the synthesis of multiple *α*-gal epitopes on glycans of viral glycoproteins produced by yeast (i.e., performing the enzymatic reaction illustrated in Figure 1A instead of the reaction in Figure 1B). *E. coli* was also engineered to serve as an expression system secreting recombinant proteins [153,154]. In addition, *E. coli* was glycoengineered to synthesize *α*-gal epitope oligosaccharides [155,156], and to synthesize N-linked glycans [154,157]. Integrating these various systems may enable the production of recombinant viral glycoproteins capped with *α*-gal epitopes to be used for vaccine preparation. Similarly, viral glycoproteins presenting *α*-gal epitopes may be produced in eukaryotic cells as in methods #1 and #2.

The examples above for producing anti-viral vaccines presenting *α*-gal epitopes are but few out of the many effective methods presently available for the preparation of such vaccines. Additional studies are required for determining the optimal methods for production, which may vary from one type of virus to the other. However, it should be noted that the direct linking of synthetic *α*-gal epitopes to amino acids (e.g., lysine) of the vaccinating proteins may alter the structure of antigens within vaccines. In some vaccines this may theoretically result in decreased efficacy of the elicited immune response because of lower recognition of the corresponding antigens on the native pathogen.

## 9. Conclusions

The carbohydrate chains (glycans) on proteins of virus envelopes form the “glycan shield”, which contributes to the survival of viruses by camouflaging the envelope protein antigens from the immune system of infected individuals. These glycans are synthesized by the glycosylation machinery of the host cells. Fortuitously, the human immune system produces natural antibodies against some antigens presented by viruses of animals (zoonotic viruses) that may be deleterious upon infection of humans. Although natural anti-carbohydrate antibodies are primarily produced against bacterial antigens, several of these antibodies also bind to carbohydrate antigens on glycan shields of zoonotic viruses. Two of these antigens are the *α*-gal epitope and Neu5Gc, binding the natural human anti-Gal and anti-Neu5Gc antibodies, respectively.

The *α*-gal epitope is synthesized on viruses infecting non-primate mammals, lemurs and New World monkeys, whereas anti-Gal is naturally produced in humans as ~1% of immunoglobulins and is found also in apes and Old World monkeys, all of which lack *α*-gal epitopes. The Neu5Gc antigen is synthesized on viruses replicating in many non-primate mammals, Old World monkeys and apes. In contrast, humans lack Neu5Gc but produce natural anti-Neu5Gc antibody. A number of observations strongly suggest that anti-Gal and anti-Neu5Gc antibodies protect humans against invading zoonotic viruses presenting *α*-gal epitopes or Neu5Gc on their glycan shields. Anti-Gal antibodies were found to destroy and neutralize viruses presenting *α*-gal epitopes and target the lysed viruses for effective uptake by APC via Fc/Fc receptor interaction, thereby increasing the immunogenicity of viral protein antigens.

The distribution of the *α*-gal epitope and the anti-Gal antibody in primates, and the identity of the base deletion mutations inactivating the *α1,3GT* gene coding the enzyme that synthesizes this carbohydrate antigen, all suggest that the *α*-gal epitope was eliminated in ancestral Old World primates 20–30 million years ago in an epidemic mediated by a lethal virus that replicated in ancestral Old World primates and presented *α*-gal epitopes on its glycan shield. Few mutated offspring that accidentally carried a base deletion mutation inactivating the *α1,3GT* gene (*GGTA1*) did not synthesize the *α*-gal epitope. Instead, these mutated offspring produced the natural anti-Gal antibody, which protected them from extinction by the virus presenting *α*-gal epitopes that killed the parental populations. A similar scenario is suggested for ancestral hominins synthesizing Neu5Gc, which were eliminated and only few mutated offspring survived. In these offspring an accidental mutation prevented Neu5Gc synthesis and led to production of protective anti-Neu5Gc antibody. These mutated offspring replaced the ancestral parental hominins synthesizing Neu5Gc.

The potency of natural anti-carbohydrate antibodies is further observed in viruses transmitted human to human. When these viruses replicate in host cells synthesizing blood group A or B antigens, they present these carbohydrate antigens on their glycan shield. The natural anti-blood group A and B antibodies in the serum of blood group O individuals readily bind to the corresponding blood group antigen on the glycan shield of such viruses and induce complement-dependent lysis of the viruses.

Based on experimental observations, it is suggested that glycoengineering of viral glycan shields to present multiple *α*-gal epitopes will markedly increase the efficacy of vaccines prepared from such viruses because of effective anti-Gal-mediated targeting of vaccines to APC. Such glycoengineering is feasible by the in vitro use of recombinant *α1,3GT*, replication of the virus in host cells engineered to have multiple copies of the *α1,3GT* gene, or by production of viral glycoproteins with N-linked glycans in yeast or bacteria expression systems that have transgenes of the corresponding glycosyltransferases. Based on data in mouse experimental models producing anti-Gal, viral vaccines presenting multiple *α*-gal epitopes are likely to be much more effective in eliciting a protective anti-virus immune response in humans than similar vaccines lacking this epitope.

## Figures and Tables

**Figure 1 ijms-21-06702-f001:**
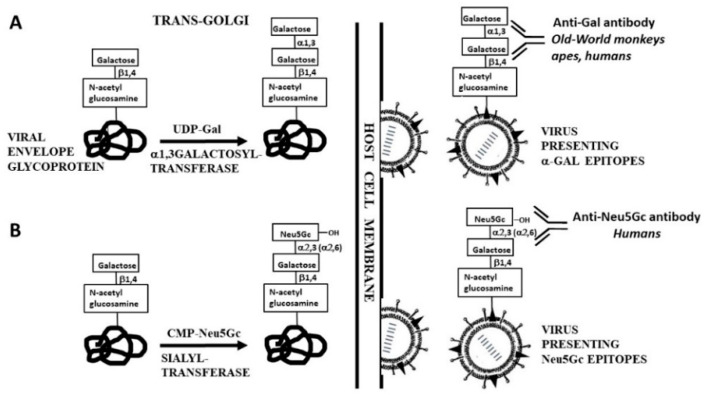
Synthesis of *α*-gal epitopes (**A**) and N-glycolyl-neuraminic acid (Neu5Gc) (**B**) on glycans of enveloped virus glycoproteins. The “capping” of the glycans occurs in the trans-Golgi compartment of host-cells and is mediated by *α*1,3galactosyltransferase and *α*2,3 and *α*2,6 sialyltransferases, using UDP-Gal and CMP-Neu5Gc as high-energy sugar donors, respectively. The hydroxyl of Neu5Gc differentiating it from Neu5Ac is indicated. Viral glycoproteins are assembled on cell membranes to form the virus envelope, followed by budding of the virus. Antibodies binding to these glycans and the species producing them are indicated. Modified from ref. [16].

**Figure 2 ijms-21-06702-f002:**
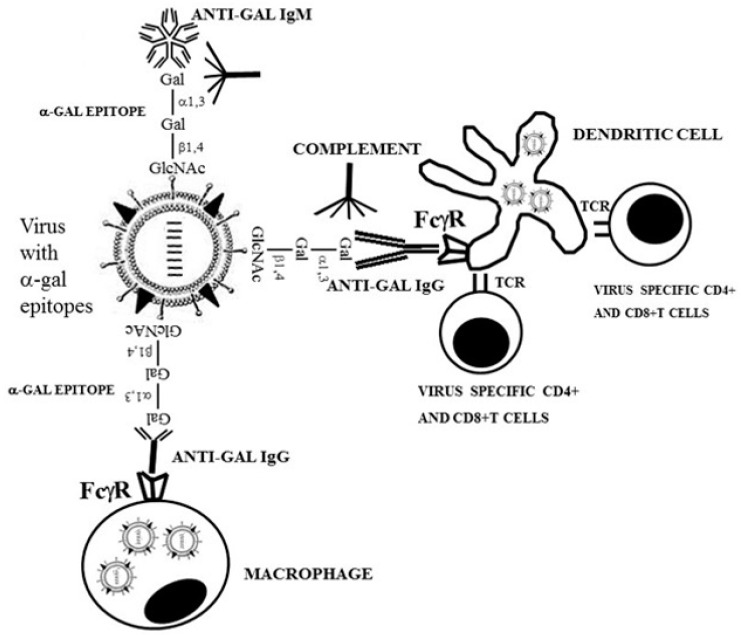
Amplifying the specific anti-virus immune response by formation of immune complexes between natural anti-carbohydrate antibodies and lysed virus, as demonstrated by the natural anti-Gal antibody. Anti-Gal binding to *α*-gal epitopes on viral glycans activates the complement cascade causing virolysis and neutralization of the virus and recruitment of antigen presenting cells (APC) such as dendritic cells and macrophages by complement cleavage peptides. Subsequent binding of the anti-Gal/virus immune complex to the Fcγ receptor (FcγR) on dendritic cells and macrophages induces extensive uptake of the opsonized virions by these APC and transport of the internalized virus to the regional lymph nodes. The APC further process and present the viral immunogenic peptides on major histocompatibility complex (MHC) molecules for the activation of CD8^+^ cytolytic T cells (CTL) and CD4^+^ helper T cells. The latter cells provide help to B cells producing virus specific neutralizing antibodies. The CTL detect various host cells infected with the virus and kill them. Modified from “The natural anti-Gal antibody as foe turned friend in medicine” by U. Galili, 2018, Elsevier/Academic Press, p. 153.

**Figure 3 ijms-21-06702-f003:**
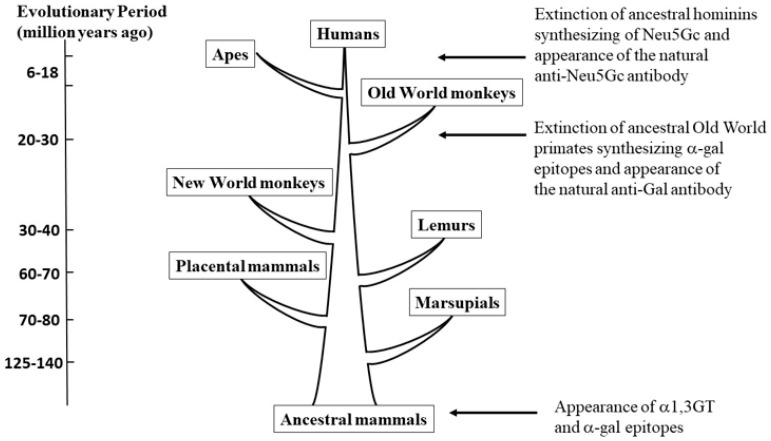
Suggested evolutionary periods (indicated by arrows) in which the *α*-gal epitope and Neu5Gc carbohydrate antigens were eliminated and the anti-Gal and anti-N-glycolyl neuraminic acid (anti-Neu5Gc) antibodies appeared as a result of viral epidemics. The presence of *α*-gal epitopes only in mammals and in no other vertebrates implies that synthesis of *α*-gal epitopes initiated in mammals at an early evolutionary period before marsupials and placental mammals diverged from a common ancestor. The absence of *α*-gal epitopes only in humans, apes and Old World monkeys implies that the elimination of primates synthesizing this epitope and the corresponding production of the natural anti-Gal antibody occurred in ancestral Old World primates (*catarrhini*) after they diverged from New World monkeys (*platirrhini*). N-5-glycolyl-neuraminic acid (Neu5Gc) is synthesized in most mammals (including apes and Old World monkeys) and other vertebrates, but it is absent in humans. In contrast, humans produce the natural anti-Neu5Gc antibody. This strongly suggests that elimination of Neu5Gc occurred after hominins and ancestors of chimpanzee diverged from a common ancestor. Modified from “The natural anti-Gal antibody as foe turned friend in medicine” by U. Galili, 2018, Elsevier/Academic Press, p. 25.

**Figure 4 ijms-21-06702-f004:**
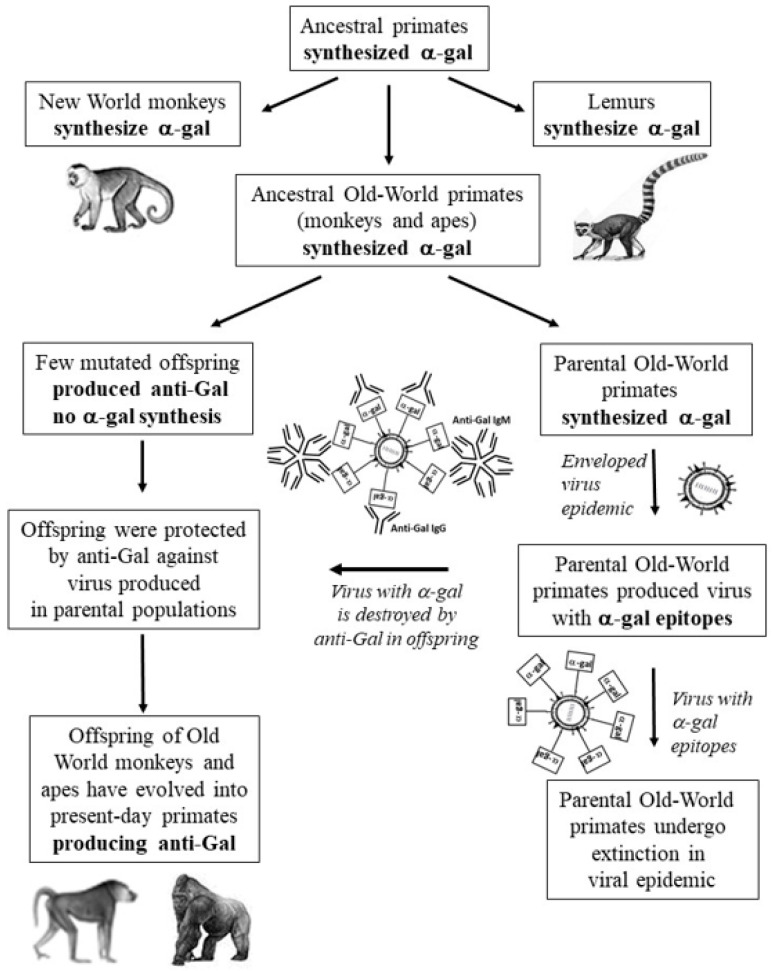
Hypothesis on the association between the evolutionary elimination of *α*-gal epitopes in ancestral Old World primates and an epidemic caused by a virus that was lethal to these ancestral primates. Early ancestral Old World primates synthesized *α*-gal epitopes, as New World monkeys, lemurs and non-primate mammals. Accidental base deletion mutation in one early Old World primate resulted in the inactivation of the *α1,3GT* gene (*GGTA1*). After few generations, a small offspring population homozygous for this inactivating mutation produced the natural anti-Gal antibody. An epidemic by a lethal virus caused the extinction of the parental populations synthesizing *α*-gal epitopes. The virus produced in parental populations presented multiple *α*-gal epitopes on the glycan shield (“*α*-gal” in the rectangles). Anti-Gal in individuals of the mutated population protected against the virus by binding to the *α*-gal epitopes on its glycan shield. This ultimately resulted in the replacement of the parental populations synthesizing *α*-gal epitopes with Old World primate progeny lacking this epitope and producing the natural anti-Gal antibody. Modified from ref. [89].

**Figure 5 ijms-21-06702-f005:**
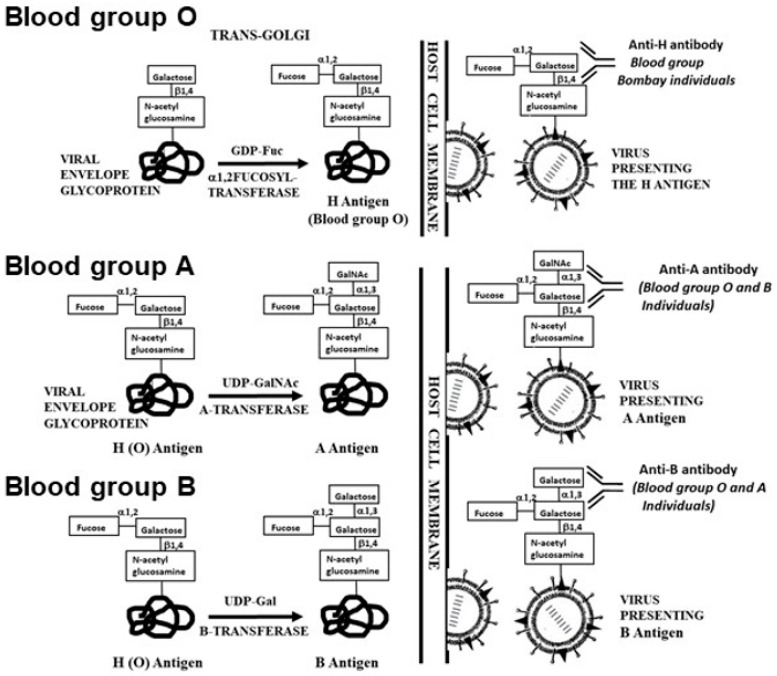
Synthesis of H (O), A and B antigens on the glycan shield of enveloped virus glycoproteins as a result of replication in human host-cells of various blood groups. The corresponding antibodies are according to the blood group. Blood group Bombay individuals have mutations that inactivate the two *α1,2FT* genes *FUT1* and *FUT2*, therefore they cannot synthesize the H antigen (blood group O) and they produce natural anti-H (anti-O) antibody in addition to the production of anti-A and anti-B antibodies. Blood group A and B antigens are synthesized on the H antigen by the corresponding A and B transferases. The synthesized carbohydrate antigens are presented on the glycan shield of the budding virus and bind natural antibodies, as indicated. Modified from ref. [16].

**Table 1 ijms-21-06702-t001:** Carbohydrate antigens on virus glycan shield recognized by human natural antibodies.

Name of Antigen	Structure of the Antigen	Species Synthesizing the Antigen	Human Antibody Binding the Antigen
*α*-gal epitope	Gal*α*1-3Gal*β*1-4GlcNAc-R	Non-primate mammals, lemurs, New-World monkeys	Anti-Gal (all humans)
Neu5Gc (N-glycolyl neuraminic acid)	Neu5Gc-R	Apes, Old-World monkeys, most non-primate mammals	Anti-Neu5Gc (all humans)
Blood group A	GalNAc*α*1-3(Fuc*α*1-2)Gal*β*1-4GlcNAc-R	Humans	Anti-A (blood group O and B individuals)
Blood group B	Gal*α*1-3(Fuc*α*1-2)Gal*β*1-4GlcNAc-R	Humans	Anti-B (blood group O and A individuals)
Blood group O (H antigen)	Fuc*α*1-2Gal*β*1-4GlcNAc-R	Humans	Anti-H (produced only in blood group Bombay individuals)

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
