# Peer review of "Host Synthesized Carbohydrate Antigens on Viral Glycoproteins as “Achilles’ Heel” of Viruses Contributing to Anti-Viral Immune Protection"

_ijms, 2020, doi:10.3390/ijms21186702_

Round 1
Reviewer 1 Report
The current review by Dr. Uri Galili refers to the protective function of carbohydrate epitopes against infection of humans by zoonotic enveloped viruses, the alpha-Gal and Neu5Gc epitopes, and the ABH blood group epitopes. The review is well-written by an expert in the field, and provides valuable information on a highly relevant aspect. However, the review could benefit from shortening some passages that are highly redundant. The principle of the protective effect is repeatedly outlined in several sections/paragraphs. Information referred to in great detail in the main body of the text is repeated in the conclusions. In particular, the latter part must be shortened and written in a more confined manner. In addition to these general remarks there are a couple of minor concerns and questions to the author, which should be considered by the author.
Abstract: It should be more clearly stated that the protective “barrier” effect in case of the alpha-Gal and Neu5Gc epitopes is solely confined to the transfer of zoonotic viruses from nonprimate mammalian reservoirs to humans, as the spreading of viruses in the human population is driven by “humanized” viruses expressing human glycoprofiles. Only in the blood group ABH context there is a more general blood group-dependent protection.
Introduction:
line 35: The sequon for protein N-glycosylation should be more exactly described: It should read: Asn-X(except Pro)-Ser/Thr (or in rare cases Asn-X-Cys).
The author refers only to N-glycans as potential carriers of the alpha-Gal and Neu5Gc epitopes. Are the two glycan epitopes exclusively found on this class of glycans? Actually, both are expressed also on O-linked chains with type 2 LacNAc and GSLs of the lacto-series.
para 3.2, line 135: Could the author refer also to IgE? Reference to group C carbohydrate allergens could be made here.
line 233: Are anti-alpha-Gal responses actually MHC-restricted?
para 4, line 341: ..this individual..
para 5, line 452: Neu5Ac
para 6.2, line 479: It may be true that blood group 0 individuals with anti- A and anti-B antibodies bebefit from some protection against the virus (for example SARS-CoV2), as is reflected in the slightly reduced risk for acquiring COVID-19 (see below), but blood group AB individuals seem not to suffer from higher risk as expected. The ABO blood group in 3,694 normal people in Wuhan displayed a percentage distribution of 32.16%, 24.90%, 9.10% and 33.84% for A, B, AB and O, respectively, while the 1,775 patients with COVID-19 from Wuhan Jinyintan Hospital showed an ABO distribution of 37.75%, 26.42%, 10.03% and 25.80% for A, B, AB and O, respectively.
line 573-582: Although some of the information given in section 7 with respect to Bombay individuals is interesting for readers, the sentences in lines 573 to 582 are purely speculative and lack any reference to evidence.
Author Response
Reviewer 1
Open Review
(x) I would not like to sign my review report
( ) I would like to sign my review report
English language and style
( ) Extensive editing of English language and style required
( ) Moderate English changes required
( ) English language and style are fine/minor spell check required
(x) I don't feel qualified to judge about the English language and style
|
Is the work a significant contribution to the field? |
|
|
Is the work well organized and comprehensively described? |
|
|
Is the work scientifically sound and not misleading? |
|
|
Are there appropriate and adequate references to related and previous work? |
|
|
Is the English used correct and readable? |
Comments and Suggestions for Authors
The current review by Dr. Uri Galili refers to the protective function of carbohydrate epitopes against infection of humans by zoonotic enveloped viruses, the alpha-Gal and Neu5Gc epitopes, and the ABH blood group epitopes. The review is well-written by an expert in the field, and provides valuable information on a highly relevant aspect. However, the review could benefit from shortening some passages that are highly redundant. The principle of the protective effect is repeatedly outlined in several sections/paragraphs. Information referred to in great detail in the main body of the text is repeated in the conclusions. In particular, the latter part must be shortened and written in a more confined manner. In addition to these general remarks there are a couple of minor concerns and questions to the author, which should be considered by the author.
Response- The Conclusions section was revised to be shorter and redundant text was deleted, as suggested.
Abstract: It should be more clearly stated that the protective “barrier” effect in case of the alpha-Gal and Neu5Gc epitopes is solely confined to the transfer of zoonotic viruses from nonprimate mammalian reservoirs to humans, as the spreading of viruses in the human population is driven by “humanized” viruses expressing human glycoprofiles. Only in the blood group ABH context there is a more general blood group-dependent protection.
Response- The suggested comment was added in lines 27,28.
Introduction:
line 35: The sequon for protein N-glycosylation should be more exactly described: It should read: Asn-X(except Pro)-Ser/Thr (or in rare cases Asn-X-Cys).
Response- Added in line 38.
The author refers only to N-glycans as potential carriers of the alpha-Gal and Neu5Gc epitopes. Are the two glycan epitopes exclusively found on this class of glycans? Actually, both are expressed also on O-linked chains with type 2 LacNAc and GSLs of the lacto-series.
Response- Added in line 42.
para 3.2, line 135: Could the author refer also to IgE? Reference to group C carbohydrate allergens could be made here.
Response- Added in lines 151-1153.
line 233: Are anti-alpha-Gal responses actually MHC-restricted?
Response- As indicated in the added text in lines 265-266, once the internalized viruses are within the endosomes of APC, the glycans including alpha-gal epitopes are destroyed by glycosidases and the processing and presentation of antigenic peptides on MHC of the APC is a natural activity of APC and not associated with any particular MHC.
para 4, line 341: ..this individual..
Response- Corrected in line 381. Thanks.
para 5, line 452: Neu5Ac
Response- Corrected in line 507. Thanks.
para 6.2, line 479: It may be true that blood group 0 individuals with anti- A and anti-B antibodies bebefit from some protection against the virus (for example SARS-CoV2), as is reflected in the slightly reduced risk for acquiring COVID-19 (see below), but blood group AB individuals seem not to suffer from higher risk as expected. The ABO blood group in 3,694 normal people in Wuhan displayed a percentage distribution of 32.16%, 24.90%, 9.10% and 33.84% for A, B, AB and O, respectively, while the 1,775 patients with COVID-19 from Wuhan Jinyintan Hospital showed an ABO distribution of 37.75%, 26.42%, 10.03% and 25.80% for A, B, AB and O, respectively.
Response- Text deleted in line 536 – “and anti-blood group B”
line 573-582: Although some of the information given in section 7 with respect to Bombay individuals is interesting for readers, the sentences in lines 573 to 582 are purely speculative and lack any reference to evidence.
Response- Text was deleted in lines 645 to 647. The term “one may hypothesize” in line 638 was changed to “one may speculate”. This manuscript is a review based on many previous published studies of my group and other groups. Based on all these studies, it is legitimate to speculate regarding future scenarios. Therefore, the change of the term “hypothesize” to “speculate”.
Submission Date
16 August 2020
Date of this review
03 Sep 2020 13:53:02
Reviewer 2 Report
This a very well written and exciting review covering the field of the protection of naturally occurring antibodies directed against human non-self carbohydrate epitopes of zoonotic viruses covering a life-time long achievement of the author and reaching into the ongoing Covid-19 pandemic. Specific focus is given to the anti-Gal antibodies, directed towards the alfa-gal epitope (Gala1-3Gal) in which Dr Galili has been the leading scientist, reaches over to the anti-NeuGc antibodies and also covering some aspects of anti-blood group AB(H) antibodies. The text is fluent and except for only a very small number of misprints, is very easy to read and comprehend. The references are highly relevant and covers a long time period from 1952 (Bhende, Y.M.; Deshpande, C.K.; Bhatia, H.M.; Sanger, R.; Race, R.R.; Morgan, W.T.; Watkins, W.M. A new 964 blood group character related to the ABO system. Lancet 1952, 1, 903-904. ) to 2020 ( Galili, U. Amplifying immunogenicity of prospective Covid-19 vaccines by glycoengineering the 1066 coronavirus glycan-shield to present a-gal epitopes. Vaccine 2020, In Press.) covering almost 60 (sic!) years of active research.
The concepts reviewed are not novel, as is well referenced in the manuscript, but their consequences are nicely and adequately formulated when applied to present and future formulations of vaccines or vaccine strategies against zoonotic membrane viruses, i.e. the SARS-Cov 2 virus and other forthcoming viruses that threatens to eradicate the majority of the human population in “catastrophic selections” (Lewis, H. Catastrophic selection as a factor in speciation. Evolution 1962, 16, 257-271.) I would however like to stress a few, not ordered, issues that the author and the editor perhaps would like to reflect upon.
- The description of the Bombay pheno- and genotype/s occurring in several paragraphs is unfortunately incomplete since humans have two a1,2-fucosyltransferase genes (FUT1 and FUT2) which should be clarified in the text. Knocking out both genes gives the true Bombay phenotype (which is probably what the author is referring to), knocking out only FUT1 gives the para-Bombay phenotype and knocking out only FUT2 gives the non-secretor phenotype. The former two kinds of individuals are very rare but the non-secretors are around 20% of the world population and thus very common! Please correct!
- In conjunction with the point above the legend of Figure 1 refers to sialyltransferases in singular. Again there are several sialyltransferases that could be accounted for being responsible for alpha 2,3; 2,6 or 2;8 sialylations to Gals or GalNAcs of different glycan core structures. Please correct.
- The effects of different isotypes of the anti-carbohydrate antibodies should be clarified in the text (and not only schematically in the figures and on line 135) since those isotypes (i.e. IgG, IgA, IgM) will have different effects on complement activation and virolysis, agglutinating viral particles or interacting with APC. Please clarify more distinctly.
- Concerning the PERV-story, described on line 210 and REF 56, – was there any evidence for that the Anti-Gal antibodies were involved in the protection against porcine RNA viruses? Please clarify.
- The word “killing” in terms of viruses is misleading since viruses are not living organisms. Replace with lysing?
- It would be interesting to have the literature view on how carbohydrate antigens at the molecular level elicits an immune response causing the "natural antibodies" to occur both in the newborn to adult perspective but also in the hominin to homo sapiens sapiens perspective. Any comments?
- It is of course difficult to prove the importance of the different host genetic mutations taking place over 25 million years ago. However, I do not agree with the author writing on page 12 that “For obvious reasons it is not feasible to measure in humans the extent of immune protection by natural anti-Gal and anti-Neu5Gc against infections of zoonotic viruses.” If the author is referring to experimental situations I do agree but if you consider the recent zoonotic viruses like HIV, SARS, MERS, Ebola and SARS2 there is a lot of both genetics and serology to be tested. Rephrase the sentence please.
- Since the virolysis is dependent on complement activation it would be interesting to refer to the development of the complement factors over the last 200 million years of mammalian development. Is there any evidence for a gain of function that would potentiate the protective effect of naturally occurring anti-carbohydrate antibodies?
- The ongoing pandemic is driving science in an unprecedented speed so I would like the author to include in this review another recent publication about the protective effect of naturally occurring anti-carbohydrate antibodies (Breiman A, Ruvën-Clouet N, Le Pendu J. Harnessing the natural anti-glycan immune response to limit the transmission of enveloped viruses such as SARS-CoV-2. PLoS Pathog. 2020;16(5):e1008556. Published 2020 May 21. doi:10.1371/journal.ppat.1008556).
- On lines 533- the author writes “This elimination of blood group A individuals by lethal viruses and the resistance of blood group O individuals due to production of natural anti- blood group A antibody that destroys virus transmitted from blood group A individuals may further support the above suggested scenario of virus mediated evolutionary elimination of ancestral Old World primates synthesizing the -gal epitope and of hominins synthesizing Neu5Gc.” I think that one hypothesis, irrespective of how strong the indications for it being true actually are, cannot support another hypothesis but maybe "be in line with" the other hypothesis. Note, I am not evaluating the hypotheses in this statement. Please consider rephrasing.
Author Response
Reviewer 2
Open Review
(x) I would not like to sign my review report
( ) I would like to sign my review report
English language and style
( ) Extensive editing of English language and style required
( ) Moderate English changes required
(x) English language and style are fine/minor spell check required
( ) I don't feel qualified to judge about the English language and style
|
Is the work a significant contribution to the field? |
|
|
Is the work well organized and comprehensively described? |
|
|
Is the work scientifically sound and not misleading? |
|
|
Are there appropriate and adequate references to related and previous work? |
|
|
Is the English used correct and readable? |
Comments and Suggestions for Authors
This a very well written and exciting review covering the field of the protection of naturally occurring antibodies directed against human non-self carbohydrate epitopes of zoonotic viruses covering a life-time long achievement of the author and reaching into the ongoing Covid-19 pandemic. Specific focus is given to the anti-Gal antibodies, directed towards the alfa-gal epitope (Gala1-3Gal) in which Dr Galili has been the leading scientist, reaches over to the anti-NeuGc antibodies and also covering some aspects of anti-blood group AB(H) antibodies. The text is fluent and except for only a very small number of misprints, is very easy to read and comprehend. The references are highly relevant and covers a long time period from 1952 (Bhende, Y.M.; Deshpande, C.K.; Bhatia, H.M.; Sanger, R.; Race, R.R.; Morgan, W.T.; Watkins, W.M. A new 964 blood group character related to the ABO system. Lancet 1952, 1, 903-904. ) to 2020 ( Galili, U. Amplifying immunogenicity of prospective Covid-19 vaccines by glycoengineering the 1066 coronavirus glycan-shield to present a-gal epitopes. Vaccine 2020, In Press.) covering almost 60 (sic!) years of active research.
The concepts reviewed are not novel, as is well referenced in the manuscript, but their consequences are nicely and adequately formulated when applied to present and future formulations of vaccines or vaccine strategies against zoonotic membrane viruses, i.e. the SARS-Cov 2 virus and other forthcoming viruses that threatens to eradicate the majority of the human population in “catastrophic selections” (Lewis, H. Catastrophic selection as a factor in speciation. Evolution 1962, 16, 257-271.) I would however like to stress a few, not ordered, issues that the author and the editor perhaps would like to reflect upon.
- The description of the Bombay pheno- and genotype/s occurring in several paragraphs is unfortunately incomplete since humans have two a1,2-fucosyltransferase genes (FUT1 and FUT2) which should be clarified in the text. Knocking out both genes gives the true Bombay phenotype (which is probably what the author is referring to), knocking out only FUT1 gives the para-Bombay phenotype and knocking out only FUT2 gives the non-secretor phenotype. The former two kinds of individuals are very rare but the non-secretors are around 20% of the world population and thus very common! Please correct!
Response- Text added in lines 550, 613–616 and 621, indicating the inactivation of both FUT1 and FUT2 genes in Bombay individuals.
- In conjunction with the point above the legend of Figure 1 refers to sialyltransferases in singular. Again there are several sialyltransferases that could be accounted for being responsible for alpha 2,3; 2,6 or 2;8 sialylations to Gals or GalNAcs of different glycan core structures. Please correct.
Response- The sialyltransferases (a2,3 and a2,6) are indicated in the legend of Figure 1, line 120 and in the text mentioning “sialyltransferases” instead of “sialyltransferases”.
- The effects of different isotypes of the anti-carbohydrate antibodies should be clarified in the text (and not only schematically in the figures and on line 135) since those isotypes (i.e. IgG, IgA, IgM) will have different effects on complement activation and virolysis, agglutinating viral particles or interacting with APC. Please clarify more distinctly.
Response- Text detailing the function of the various isotypes is added in lines 147-153.
- Concerning the PERV-story, described on line 210 and REF 56, – was there any evidence for that the Anti-Gal antibodies were involved in the protection against porcine RNA viruses? Please clarify.
Response- Text clarifying the possible protection by anti-Gal against viruses presenting alpha-gal epitopes is added in lines 242-245.
- The word “killing” in terms of viruses is misleading since viruses are not living organisms. Replace with lysing?
Response- The word “killing” was replaced with the word ”lysis” and the word “killed”, in association with virus was replaced with “lysed” throughout the manuscript.
- It would be interesting to have the literature view on how carbohydrate antigens at the molecular level elicits an immune response causing the "natural antibodies" to occur both in the newborn to adult perspective but also in the hominin to homo sapiens sapiens perspective. Any comments?
Response- Text on natural anti-Gal produced in newborns at the molecular level production is added in lines 166-176. Regretfully, I did not find information related to natural antibodies associated with hominin to homo sapiens sapiens perspective.
- It is of course difficult to prove the importance of the different host genetic mutations taking place over 25 million years ago. However, I do not agree with the author writing on page 12 that “For obvious reasons it is not feasible to measure in humans the extent of immune protection by natural anti-Gal and anti-Neu5Gc against infections of zoonotic viruses.” If the author is referring to experimental situations I do agree but if you consider the recent zoonotic viruses like HIV, SARS, MERS, Ebola and SARS2 there is a lot of both genetics and serology to be tested. Rephrase the sentence please.
Response- The sentence in line 515 was deleted and the next sentence in line 516 was rephrased.
- Since the virolysis is dependent on complement activation it would be interesting to refer to the development of the complement factors over the last 200 million years of mammalian development. Is there any evidence for a gain of function that would potentiate the protective effect of naturally occurring anti-carbohydrate antibodies?
Response- Regretfully, I could not find scientific information related to evolution of the complement system in vertebrates which may be related to the manuscript.
- The ongoing pandemic is driving science in an unprecedented speed so I would like the author to include in this review another recent publication about the protective effect of naturally occurring anti-carbohydrate antibodies (Breiman A, Ruvën-Clouet N, Le Pendu J. Harnessing the natural anti-glycan immune response to limit the transmission of enveloped viruses such as SARS-CoV-2. PLoS Pathog. 2020;16(5):e1008556. Published 2020 May 21. doi:10.1371/journal.ppat.1008556).
Response- Thank you for bringing up this relevant manuscript. I was not aware of it since it was published when I submitted the attached manuscript. I included the reference to this manuscript as ref. 78 and cited it in lines 324 and 475.
- On lines 533- the author writes “This elimination of blood group A individuals by lethal viruses and the resistance of blood group O individuals due to production of natural anti- blood group A antibody that destroys virus transmitted from blood group A individuals may further support the above suggested scenario of virus mediated evolutionary elimination of ancestral Old World primates synthesizing the a-gal epitope and of hominins synthesizing Neu5Gc.” I think that one hypothesis, irrespective of how strong the indications for it being true actually are, cannot support another hypothesis but maybe "be in line with" the other hypothesis. Note, I am not evaluating the hypotheses in this statement. Please consider rephrasing.
Response- The statement was rephrased as suggested 598.
Submission Date
16 August 2020
Date of this review
29 Aug 2020 10:59:34